# Tracing Acid-Base Variables in Exercising Horses: Effects of Pre-Loading Oral Electrolytes

**DOI:** 10.3390/ani13010073

**Published:** 2022-12-24

**Authors:** Amanda P. Waller, Michael I. Lindinger

**Affiliations:** 1Center for Clinical & Translational Research, Nationwide Children’s Hospital, Columbus, OH 43205, USA; 2Research and Development, The Nutraceutical Alliance Inc., Guelph, ON N1E 2G7, Canada

**Keywords:** electrolyte supplement, supplementation, fluid balance, electrolyte balance, exercise performance, recovery, physicochemical acid-base

## Abstract

**Simple Summary:**

Exercise results in changes in blood acid-base status that are proportional to the duration and intensity of the activity. Prolonged activity and transport are associated with substantial losses of water and electrolyte (ions) from the body, mostly through sweating. Because these losses can be large it is recommended that horses drink electrolyte solutions designed to replace lost water and ions. In this study horses were given either 1, 3 or 8 L of water or an oral electrolyte supplement and effects on acid-base state were measured. It was found that small volumes (1 and 3 L) had minimal or no effect. The 8 L of solution was designed to fully replace electrolyte and water losses. The large volume (8 L) electrolyte supplement, compared to water, abolished the mild alkalosis (raised pH) that occurred with long-lasting submaximal exercise.

**Abstract:**

Oral electrolyte supplementation may influence acid-base state during exercise due to the intestinal absorption of administered water and electrolytes used to mitigating sweat losses. This study examined the effect of pre-exercise electrolyte supplementation (3 and 8 L) on plasma acid-base variables at rest, during moderate intensity exercise and during recovery. It was hypothesized that electrolyte supplementation will result in improved acid-base state compared to the alkalosis typical of prolonged exercise. In randomized crossover fashion, four horses were administered 3 L or 8 L of a hypotonic electrolyte solution (PNW) intended to replace sweat losses, or water alone (CON), 1 h before treadmill exercise to fatigue (at 35% of peak VO_2_) or for 45 min at 50% peak VO_2_. Blood was sampled at 10-min intervals before, during and after exercise, and analyzed for dependent and independent acid-base variables. Effects of 3 L of supplementation at low exercise intensities were minimal. In the 8 L trials, plasma [H^+^] decreased (*p* < 0.05) during exercise and early recovery in CON but not PNW. Plasma TCO_2_ decreased (*p* < 0.05) by 30 min after PNW reaching a nadir of 28.0 ± 1.5 mmol/L during the early exercise period (*p* = 0.018). Plasma pCO_2_ and strong ion difference [SID] were the primary contributors to changes in [H^+^] and [TCO_2_], respectively. Pre-exercise PNW of 8 L intended to fully replenish sweat loses maintained [H^+^], decreased [TCO_2_] and mitigated the mild alkalosis during moderate intensity exercise.

## 1. Introduction

Significant acid-base disturbances are a consequence of moderate to high intensity exercise [1], as well as prolonged duration exercise [2,3], in horses and can be associated with various pathologies including dehydration, hypochloremic alkalosis, and renal failure [4]. Additionally, there is strong interest in acid-base balance within the context of athletic performance due to the association between acidification and muscle fatigue [5,6]. Knowledge of the origins of acid-base disturbance is needed in order to understand basic, applied and clinical physiology. While acid–base state has traditionally been described by only pH, bicarbonate concentration ([HCO_3_^−^]), and plasma partial pressure of carbon dioxide (pCO_2_), this definition does not identify the physiological inputs that determine and contribute to altered acid–base state [4,7]. Simply, the traditional acid-base variables do not provide a mechanistic basis for understanding acid-base physiology.

Identification of the origins of (independent factors contributing to) acid-base disturbances is accomplished by use of the physicochemical approach to acid-base status [4,8,9], which recognizes underlying physical and chemical principles demonstrating that the dependent variables are determined by a change in one or more of the independent variables (Figure 1). The independent variables that determine acid–base status are the pCO_2_, the strong ion difference ([SID]; concentrations of strong ions in plasma) and [Atot] (mainly the weak acid charge of plasma albumin). Because of these relationships it is recognized that both exercise-induced losses of ions (through sweating) and dietary consumption of ionic minerals (electrolytes) have direct, acute [7] and/or chronic [10,11] effects on plasma acid–base balance.

There are few studies investigating the effects of electrolyte supplementation on equine acid–base balance at rest [10,12] or with exercise [11,13,14]. For example, adding 50–100 g per day of NaCl to the diet of exercising horses resulted in systemic acidification with increased renal acid excretion after 2 weeks of supplementation [15]. While electrolyte pre-loading prior to exercise benefits both performance and recovery [3,16,17] few studies have investigated the effects of pre-exercise electrolyte supplementation on acid-base balance. Hess and coworkers studied the effects of administration of K^+^-containing electrolyte mixtures and K^+^-free electrolyte mixtures that were administered orally by syringe 1–2 h before the ride and at approximately 20 km intervals during an 80 km endurance ride [18]. The amounts of electrolytes administered were low, and water intake was not adequate to replace sweat fluid losses as horses had increased concentrations of plasma ions, osmolality, proteins with increasing distance, together with decreased plasma [H^+^] and [Cl^−^]. In another study, researchers determined that administration of seven different commercial electrolyte supplements as hypertonic pastes prior to a simulated race test had no effect on plasma concentrations of electrolytes, TCO_2_ or plasma protein concentration [19]. There were several limitations of this study including the use of pastes, the quantity of electrolytes given were low (approximating a 2 L sweat loss), the horses were unfit, and the time course for analysis of acid-base responses was limited to only 60 and 90 min post-exercise.

The present paper represents the final part of an extensive series of experiments that traced oral electrolytes in resting and exercising horses, from gastric emptying and intestinal absorption through to their appearance in plasma and ultimately uptake into cells and tissues [17,20,21]. The purpose of the present study was to employ the physicochemical acid-base approach to provide a comprehensive time-course of the factors contributing to the acid-base alterations that occur following pre-exercise electrolyte supplementation in athletic horses, with frequent blood samples taken before, during, and after two intensities of exercise. We hypothesized that pre-loading (administration of electrolyte supplements pre-exercise) with an amount of water and electrolytes (using a balanced oral electrolyte solution) sufficient to replace predicted sweat losses, will attenuate the acid-base disturbances that occur during submaximal exercise and during recovery. 

## 2. Materials and Methods

### 2.1. Animals

Horses in this study were part of the University’s research herd. The care and use of animals for this research were approved by the University of Guelph Animal Care Committee, and the experiments were conducted in compliance with the University’s animal care policy and in accordance with the guidelines of the Canadian Council on Animal Care.

During the 6–8 week period of exercise conditioning and 6–8 week period of the study horses received grass hay and water ad libitum (approximately 2.5 kg of hay). In two meals daily, horses also received 2.5 kg/day grain and mineral mixture (Purina Checkers: 14% protein, 6% fat, 10% fibre. Details of the rations are provided in the previous publication [17]. The estimated DCAD (Na^+^ + K^+^ − 2Cl^−^) for the combined hay plus grain was 274 mEq/kg dry matter. Horses had access to a 3–acre paddock, with very minimal forage, during the day.

The same four mares (Standardbred, Thoroughbred and crosses, 5–12 years of age, 425–500 kg) were used for the six experimental trials. Prior to study commencement the horses were sedentary research horses that had not been used for experiments for a minimum of 3 months. The horses were progressively exercise conditioned using an indoor equine treadmill for 6–8 weeks. A minimum of 2 weeks separated the 3 trials. All horses remained healthy throughout the trials. There were no adverse effects due to the research protocols.

### 2.2. Study Design

The overall study design consisted of two experimental series with different exercise protocols, each performed as a set of crossover trials with varying pre-exercise electrolyte treatments (see Table 1). Series 1 consisted of a prolonged low-intensity (35% peak VO_2_; medium trot) exercise test to voluntary fatigue (unable to keep to the speed of the treadmill) with two treatment trials performed in randomized crossover design comparing (a) pre-loading with 1 L of water alone (H_2_O-1) as a control with (b) pre-loading with 3 L of electrolyte supplement (PNW-3). Series 2 consisted of a moderate intensity (50% peak VO_2_; fast trot) exercise test with a set duration of 45 min with two treatment trials performed in randomized crossover design comparing the effects of pre-loading with 8 L of electrolyte supplement (PNW-8) or 8 L of water alone (H_2_O-8).

### 2.3. Experimental Procedures

The experiments were conducted in the morning. Nasogastric administration of the fluid/electrolyte treatments occurred 3 h after the 07:00 morning feeding, and exercise commenced 1 h later. Preparation of the horses for the trial was as follows. The hair coat over the jugular vein, 10–20 cm below the mandible, was clipped on both sides of the neck. A topical analgesic [EMLA cream (2.5% lidocaine and 2.5% prilocaine), Astra Pharma, Mississauga, ON, Canada] was applied to desensitize the skin 30–45 min before insertion of catheters. Both right and left catheterization sites were aseptically prepared for catheter insertion. Local anesthetic (2% Xylocaine, Astra Pharma, Mississauga, ON, Canada) was injected subcutaneously to complete the analgesia. Catheters (14-gauge, 5.25-in. Angiocath, Becton-Dickinson, Mississauga, ON, Canada) were inserted anterograde, secured with tape and stitches to the skin. 20-inch extensions with 4-way stopcocks (Medex Inc., Hilliard, OH, USA) were attached to the catheters for ease of blood sampling. Patency of the catheters was maintained with sterile, heparinized 0.9% NaCl (2000 IU/L NaCl).

A powdered oral electrolyte supplement (PNW, Perform’N Win, Buckeye Nutrition, Dalton, OH, USA) was used and consisted of a balanced formulation of sodium and potassium chloride, magnesium sulfate, dextrose, sucrose, calcium citrate, fumaric acid and citric acid. The PNW was designed to mimic the concentrations of the major electrolytes lost in sweat. When dissolved in water according to manufacturer’s instructions, the measured electrolyte concentrations (Nova Statprofile 5, Nova Biomedical, Waltham, MA, USA) are given in Table 2. The fluid and electrolyte supplementation volumes utilized in each series were intended to fully replenish estimated exercise-induced sweat fluid and ion losses.

Exercise was performed indoors under constant, neutral ambient conditions (temperature ~22 °C and relative humidity ~55%) on a high-speed equine treadmill (Säto, Uppsala, Sweden). Cooling was provided by a large high-speed fan blowing air at the front of the horse. After an initial 5-min walk the treadmill speed was increased to 9 km h^−1^ (2.5 m s^−1^) and an incline of 1.5 degrees. The treadmill speed was then set to that speed corresponding to each horse’s conditioning as determined during the previous week of training. The exercise intensities approximated 35% and 50% of peak VO_2_ for Series 1 and Series 2, respectively. In Series 1, the average speed was 3 m/s (moderate trot) maintained to the point of voluntary fatigue (inability to keep up with the speed of the belt). In Series 2, average treadmill speed was 4 m/s (fast trot) for a set time of 45 min. 

### 2.4. Sample Collection

Jugular venous blood was sampled at rest before loading with electrolyte solution, following fluid/electrolyte loading, during the exercise period, and for the initial two hours of recovery. Food and water were withheld throughout the sampling period. Blood was collected into lithium heparinized vacutainer tubes and analyzed within 3 min of sampling. Measures inclued hematocrit, pH, pCO_2_, pO_2_ and the plasma concentrations of Na^+^, Cl^−^, K^+^, lactate^-^ and normalized (to pH 7.4) [Ca^2+^] using ion-selective electrodes (Nova Statprofile 5+, Nova Biomedical, Waltham, MA, USA). Plasma [HCO_3_^−^] and [TCO_2_] were calculated by the Nova Statprofile using the Henderson–Hasselbach equation. Plasma was obtained from remaining blood by centrifugation at 15,000 g for 5 min. Plasma protein concentration ([PP]) was measured using a clinical refractometer (Atago clinical refractometer model SPR-T2; Atago, Tokyo, Japan). 

### 2.5. Analytical Methods and Calculations

Plasma [H^+^] was calculated as the inverse log_10_ of measured pH. 

Plasma [SID] (strong ion difference) was calculated as the sum of the concentrations of the strong cations minus the strong anions [9]: [SID] = [Na^+^] + [K^+^] − [Cl^−^] − [Lactate^−^]

The plasma total weak acid concentration ([Atot]) was calculated by multiplying the [PP] (in g/dL) by 2.24 [22].

The dependent acid–base parameters (pH, [H^+^], TCO_2_, [HCO_3_^−^]) were calculated using AcidBasics II software (©2003, PD Watson, Columbia, SC, USA) as described previously [1,23]. The contributions of each of the independent variables ([SID], [Atot] and pCO_2_) to the dependent variables ([H^+^] and TCO_2_) were determined by maintaining 2 of the 3 independent variables constant and calculating [H^+^] and TCO_2_ in response to changes in the third independent variable.

### 2.6. Statistics

Statistical analysis was performed using SigmaStat software (Systat, San Jose, CA, USA). Data are presented as mean ± standard error. The data from the low- and moderate- intensity exercise series of were analyzed separately. Data were assessed using 2-way repeated measured ANOVA with respect to trial and time. When a significant F ratio was obtained, a one-way repeated measures ANOVA was used to compare means among time points. The Bonferoni post hoc test was used to test for differences among means. Significance was accepted at *p* ≤ 0.05 at a power ≥ 0.8.

## 3. Results

### 3.1. Fluid & Electrolyte Losses and Replenishment

In Series 1, the cumulative losses of water, Na^+^ and Cl^−^ in the PNW-3 trials were greater (*p* < 0.05) than in the H_2_O-1 trial (Table 3, top). The % of lost water and ions replaced in the PNW-3 trial was greater (*p* < 0.05) than in the H_2_O-1 trial.

In Series 2, the cumulative sweat fluid and ion losses were greater (*p* < 0.05) than in Series 1 (H_2_O-1; PNW-3), reflecting the increased intensity of exercise in Series 2 (Table 3, bottom). The % of lost water and ions replaced in the PNW-8 trial was greater (*p* < 0.05) than in the H_2_O-8 trial. Pre-loading with large volume electrolytes (PNW-8) was effective in mitigating exercise-induced sweat fluid and ion losses, such that these horses exhibited greater cumulative ion losses (*p* < 0.05) but lower total body ion deficits (*p* < 0.05) compared to the H_2_O-8 trial.

### 3.2. Series 1 Acid-Base Variables

The time courses for the independent acid-base variables are shown in Figure 2. There were no differences in [SID] (Figure 2A), pCO_2_ (Figure 2B), or [Atot] (Figure 2C) between trials. Compared to baseline resting samples, low intensity exercise to fatigue resulted in an increased [Atot] (*p* < 0.025) during the recovery period of both trials, as a result of increased [PP].

The time courses for the dependent acid-base variables are shown in Figure 3. There were no differences in [H^+^] (Figure 3A) or TCO_2_ (Figure 3B) between trials.

### 3.3. Series 2 Acid-Base Variables

The time courses for the independent acid-base variables are shown in Figure 4, with the time courses for each of the main strong ions shown in Figure 5. Compared to H_2_O-8, horses in PNW-8 had a decreased [SID] (*p* < 0.046) during exercise (Figure 4A), due primarily to an increased [Cl^−^] (*p* = 0.01(Figure 5 B). Plasma pCO_2_ (Figure 4B) was decreased (*p* < 0.038) during exercise in both trials. Plasma [Atot] (Figure 4C) was increased (*p* = 0.047) from baseline at the end of exercise in the H_2_O-8 trial only.

Plasma [Na^+^] (Figure 5A) and [Cl^−^] (Figure 5B) were increased (*p* = 0.036 and *p* < 0.021, respectively) after 8 L of electrolyte supplementation compared to only 8 L of water. Plasma [K^+^] (Figure 5C) was increased (*p* < 0.039) during exercise with no difference between trials. There were no differences in [lactate^-^] (Figure 5D) over time or between trials.

The time courses of the dependent acid-base variables are shown in Figure 6. Compared to PNW-8 horses, horses in the H_2_O-8 trial were alkalotic and exhibited decreased [H^+^] (Figure 6A; *p* < 0.025) and increased TCO_2_ (Figure 6B; *p* < 0.039) throughout the exercise and recovery periods. As expected, plasma [HCO_3_^−^] (not shown had a similar time course to TCO_2_, such that it was decreased (*p* < 0.024) post supplementation and during early exercise in the PNW-8 trial.

The contributions of the independent variables to the change in the dependent variables are shown in Figure 7. Plasma pCO_2_ was the primary contributor to changes in [H^+^] (Figure 7A), with a minor contribution from [Atot]. The [SID] was the primary contributor to changes in [TCO_2_] (Figure 7B), with a minor contribution from the pCO_2_. Forward stepwise regression was performed (Figure 8) and the order of entry of variables into the regression model and partial R^2^ values confirmed the relative importance and contribution of each independent variable to the changes in [H^+^] (Figure 8A) and TCO_2_ (Figure 8B)

Additional measured parameters of interest are presented in Appendix A. Hematocrit was increased (*p* < 0.048) during the exercise period in both trials (Appendix A). There were no differences in the partial pressure of oxygen (pO2; Appendix A) over time or between trials. Plasma glucose (Appendix A) was increased (*p* = 0.022) 10 min after fluid and electrolyte administration in the PNW-8 trial, and remained increased (*p* < 0.015) through the early exercise period. Ionized calcium ([iCa^2+^]; Appendix A) was decreased (*p* < 0.048) from baseline throughout the exercise and recovery periods in the H_2_O-8 trial.

Linear regression analysis showed significant positive correlations (*p* < 0.001) between the measured and calculated concentrations of independent acid-base variables [H^+^] (Appendix A) and TCO_2_ (Appendix A).

Appendix A depicts the total body losses of fluid and electrolytes. Electrolyte deficits remaining were greater in the H_2_O-8 trial compared to the other trials (*p* < 0.036). While there were greater total body losses of electrolytes in the PNW-8 trial compared to H_2_O-8 (*p* < 0.05), the electrolyte deficit remaining was decreased (*p* < 0.05) as a result of the additional electrolytes supplemented. Total fluid deficit was not different between trials.

## 4. Discussion

The present study provides a detailed time course of acid-base parameters in response to two different intensities of submaximal exercise after first pre-loading with oral electrolytes one hour before exercise using amounts intended to replace anticipated sweat losses. Horses performing a single bout of moderate intensity exercise in thermoneutral ambient conditions exhibited a mild alkalosis during exercise and early recovery. In Series 1 trials, of low intensity to voluntary fatigue, there were only small effects of 3 L electrolyte pre-loading on acid-base parameters. In contrast, in Series 2 trials (intensity of ~50% peak VO_2_) pre-loading with a stomach-full (8 L) of a commercially available, hypotonic oral electrolyte supplement decreased TCO_2_ and ameliorated the exercise-induced perturbations in acid-base state. Overall, these results show that moderate intensity exercise results in greater water and ion losses than prolonged low-intensity exercise. Furthermore, pre-exercise electrolyte supplementation has the potential to support whole body acid-base state during periods of exercise than result in significant sweat ion losses. Given the practical feasibility of the exercise and treatment interventions, these findings are likely to be applicable to a wide range of performance horses, as these horses train and compete at moderate to high exercise intensities.

### 4.1. Independent Acid-Base Variables

Performance horses frequently undertake exercise bouts that result in a disturbance of acid-base balance [2,3,23,24,25,26,27,28,29]. The disturbances in acid-base balance can be traced to alterations in the concentrations of strong and weak ions as a result of fluid and electrolyte shifts between body fluid compartments, as well as sweating and renal losses. In addition, increases in the production of metabolic products of increased muscle activity, mainly CO_2_ and lactate^-^, also contributed to the acid-base disturbance. For horses performing prolonged exercise, including transport, it is recommended that electrolyte supplements be provided orally 1–2 h prior to the activity [17,18,23]. Oral electrolyte supplementation may, however, influence acid-base state because of the provision of large amounts of water and electrolytes. The effects of pre-loading with fluid and electrolytes to fully replenish predicted exercise-induced losses had not previously been studied. Since each independent acid-base variable may be varied in different ways and for different durations, an understanding of the source of acid-base disturbance requires the ability to identify the magnitude and time course of change in the independent acid-base variables [4,23]. In the present study, both exercise- and treatment-(electrolyte supplement) effects on the independent variables contributed to the observed acid-base alterations during both moderate intensity exercise and recovery.

In contrast to the large intercompartmental fluid shifts that result in increased plasma concentrations of the strong ions Na^+^ and Cl^-^ during high intensity exercise [1], there were no appreciable changes in these ions during moderate-intensity exercise in the present study. As expected, plasma [K^+^] was increased during the moderate intensity exercise bouts—a result of rapid rates of K^+^ loss through sarcolemmal K^+^ channels during the recovery phase of action potentials [30], while there were no significant alterations in plasma [lactate^-^]. Compared to H_2_O-8, pre-loading with 8 L of PNW resulted in sustained increases in plasma [Na^+^] and [Cl^-^] throughout of the sampling period. The increased [Cl^-^] contrasts to most studies of prolonged exercise and demonstrates the effectiveness of oral electrolyte supplementation to mitigating the hypochloremic alkalosis commonly observed in endurance exercise [2,18,31]. Compared to H_2_O-8, PNW-8 result in a lower plasma [SID] due to [Cl^−^] increased to a greater amount than [Na^+^]. The reduced [SID], in turn, was responsible in large part for the decreases in plasma [H^+^] and HCO_3_^−^] (alkalosis) compared to when horses were provided only water.

Jugular venous plasma PCO_2_ was decreased during moderate intensity exercise due to an increased alveolar ventilation as a result of muscular exercise [23,32,33]. In addition, because dextrose was present in the PNW to facilitate intestinal absorption of water and Na^+^ [17,34] an increase in skeletal muscle glycolytic activity post-supplementation may contribute to the higher plasma PCO_2_ during exercise [13,35] and early recovery in the PNW-8 trial. The dextrose concentration (31 mmol/L) of the PNW electrolyte supplement provided 248 mmoles of glucose and resulted in a significant pre-exercise glycemic response and decreased plasma [glucose] during exercise (Appendix A). Similarly, when a larger amount of glucose was fed as soluble carbohydrates 2 h before a moderate intensity exercise bout (50% peak VO_2_), resulted in higher plasma [glucose] before exercise, and in lower plasma [glucose] and higher skeletal muscle utilization of blood-borne glucose during exercise [35]. This led to overall higher carbohydrate oxidation and lower lipid oxidation during exercise, and as such proportionately more CO_2_ was produced.

Plasma [Atot] (the third independent variable) was increased at the end of exercise compared to baseline in all H_2_O trials and the PNW-3 trial but not the PNW-8 trial. The initial increase in [PP] and hence [Atot] at the onset of moderate to high intensity exercise is the result of a net shift of protein-poor fluid into both the interstitium and intracellular fluid compartment of contracting skeletal muscle [36,37,38]. However, elevated [PP] and [Atot] at the end of exercise and early recovery indicates sweating-induced dehydration (with concentration of the total extracellular fluid compartment) that was partially mitigated by pre-loading horses with large volume fluid and electrolytes.

While the results of the present study were generated using blood sampled from the jugular vein, there is some similarity to the results obtained when blood is sampled from a facial artery and from the pulmonary artery (mixed venous blood) [23]. For example, in both arterial and mixed venous blood there was no plasma acidosis when exercise was performed at an intensity of ~50% of maximal heart rate. The mild alkalosis in arterial plasma during exercise was mainly due to decreased PCO_2_ and secondarily to increased [SID]. However, in mixed venous plasma, a minimal response of [H^+^] resulted from the acidifying effects of increased PCO_2_ being offset by the alkalizing effects of increased [SID] [23].

### 4.2. Dependent Acid-Base Variables

The present study employed a comprehensive physicochemical approach to identify the origins of acid–base disturbances in exercising horses in order to better understand the extent to which the acid-base disturbance may be mitigated by electrolyte pre-loading, and/or resolved during recovery from exercise. High-intensity exercise is associated with a systemic acidosis, as a result of increased lactate^-^, pyruvate^-^ and CO_2_ production in working muscles via increases in glycolytic and respiratory metabolism [39]. In contrast, prolonged low to moderate intensity exercise has been shown to induce a plasma alkalosis, as a result of sweat electrolyte losses and progressive strong anion (Cl^−^) depletion [2,18,31,40]. In the present study, however, water alone compared to pre-loading with 8 L PNW prior to a bout of moderate intensity exercise, lowered TCO_2_ by an average of 1.2 mmol/L but raised [H^+^] by an average of 2.5 nmol/L. The primary contributor to the increased [H^+^] was the increased pCO_2_ (indicating increased muscle respiratory metabolism), with minor contribution of increased [Atot]. The primary contributor to the lowered TCO_2_ was the increase in [SID], with minor contribution of pCO_2_. Additionally, low- to moderate exercise to fatigue did not result in a volume of sweat loss which lead to significant acid-base disturbances, therefore pre-loading oral electrolytes to replace the sweat fluid & ion losses also did not significantly alter acid-base status at an intensity of ~35% of peak VO_2_.

We previously demonstrated that post-exercise administration of 8 L of a balanced hypotonic commercial electrolyte solution designed to replace all sweat losses induced by a bout of moderate intensity exercise, resulted in ~2 mmol/L decrease in plasma TCO_2_ during the recovery period [41]. As in the present study, the primary contributor to the decrease in TCO_2_ was a decreased [SID], as a result of the non-significant increases in plasma [Cl^-^] with supplementation. The current study provides additional confirmation of the effectiveness of oral electrolyte supplementation in exercising horses, and further extends this to electrolyte pre-loading as well. This is of particular importance to the equine community as conventional wisdom has led to the belief that one simply cannot effectively pre-load electrolytes, since the horse has no ability to store excess amounts. While this may be true when considering a longer timeline, the present study provides clear evidence that within a carefully coordinated acute time frame, pre-loading can indeed attenuate sweating-induced fluid and electrolyte losses. Therefore, electrolytes administered into the gastrointestinal tract may serve as a temporary reservoir that can be called upon during subsequent exercise or transportation [17].

### 4.3. Practical Implications

Quantitative interpretation of acid-base status is a necessary component of optimizing equine health and performance. It is also relevant that racing jurisdictions have rules in place to determine if horses have been administered an alkalinizing substance that may enhance racing performance [42]. In many racing jurisdictions a TCO_2_ test threshold of 37 mmol/L is used to identify horses that may have been administered a prohibited alkalinizing agent [43].Nevertheless, it is becoming increasingly recognized both scientifically and clinically that simply describing acid-base state in terms of TCO_2_, pH ([H^+^]), and [HCO_3_^−^] provides an incomplete picture; they describe the effects but not the cause of any acid-base disturbance. Though still relatively uncommon, the collection of academic equine studies that use the true physicochemical approach is steadily increasing and as its use has become more commonplace, a growing body of clinical literature suggests enhanced predictive and diagnostic utility of the physicochemical over traditional methods [[23],[44],[45],[46]. The advantage of the quantitative, physicochemical approach is the ability to determine the physical and chemical origins of acid-base disturbances. This knowledge then allows for the development and administration of effective treatment strategies for correcting an acid–base disturbance. Thus, in terms of both accuracy and clinical utility, using the physicochemical approach to determine exercise and dehydration induced acid-base disturbances truly makes the most sense.

Both the low and moderate intensity exercise protocols used in the present study are of practical significance because these intensities are routine performed during ‘long-slow-distance’ conditioning, a necessary component of early-phase equine exercise conditioning. While humans produce a very dilute hypotonic sweat, equine sweat has a tonicity similar to that of plasma [21]. Thus, even a modest exercise bout can result in substantial fluid and ion losses, and ultimately acid-base disturbances. Sweating induced dehydration such as that occurring during prolonged submaximal exercise or during prolonged transport can result in metabolic alkalosis [2,3]. Pre-loading with a balanced electrolyte solution to replace the water, Na^+^, K^+^, and Cl^−^ lost in sweat effectively corrects both the strong ion and weak ion aspects of the acid–base disturbance simultaneously. Since Cl^−^ in the administered solution balances the sum of Na^+^ and K^+^, and as K^+^ is taken up by the cells the bulk of the ingested Cl^−^ will remain in the extracellular compartment with Na^+^, serving to osmotically retain extracellular fluid and thereby lower [PP]. Conversely, replenishing with water alone is ineffective, as our current findings demonstrate, and as previously shown in humans [47].

### 4.4. Limitations

There are large inconsistencies in the scientific literature regarding the electrolyte content of administered supplements, the volume administered, the intensity and duration of exercise, and the frequency (typically inadequate) of the blood sampling time course. The present study addressed each of these factors to the extent possible within the experimental design—to study performance, water and electrolyte balance when pre-loading with a physiologically balanced electrolyte supplement one hour prior to exercise. The main limitation of this study was the low number of horses (4) in each treatment. This was somewhat mitigated by the randomized crossover design and the fact the same four horses were used in all the experiments. In addition, exercise training was held constant once the treadmill trials began and the same environmental conditions were kept from trial to trial. Secondly, in the 35% peak VO_2_ intensity exercise trials (Series 1), the time-to-fatigue protocol was intended to determine if there was a performance enhancing effect of electrolyte supplementation prior to prolonged fatiguing exercise. Indeed, the increased total amount of electrolytes and fluid administered resulted in increased exercise duration in PNW-3 compared to H_2_O-1. In the 50% peak VO_2_ intensity exercise trials (Series 2) a set exercise time was used to allow a direct comparison of supplement effectiveness to be made during the exercise and recovery periods. The design, however, prevented comparison of the acid-base effects of the supplement at low- versus moderate intensities, because the differences in intensity and duration resulted in different stresses on the horses. Additionally, to avoid confounding differences in total fluid intake between horses, they were not permitted to drink fluids the 5-h blood sampling period. It is likely that recovery may have been different if voluntary drinking was allowed, as is typical practice. Finally, acid-base and blood gas measurements were not temperature corrected. While measurement of blood-gas and acid-base parameters at a standardized 37 °C is common practice both clinically and experimentally, correction to core body temperature at the time the sample was taken is likely to improve accuracy [23], but estimates of core temperature were not made in the current study.

## 5. Conclusions

The physicochemical approach provides an essential foundation for the effective treatment of pathological acid-base disorders. A single bout of moderate intensity exercise performed in neutral ambient conditions can lead to substantial fluid and ions losses that perturb acid-base balance during the exercise and initial recovery periods. Pre-loading with hypotonic oral electrolytes to fully replenish sweat losses ameliorates the dehydration-induced alkalosis and decreases plasma TCO_2_. Finally, effects of feed and electrolyte supplementation should be considered when interpreting electrolyte, plasma protein, and acid-base results of any blood test.

## Figures and Tables

**Figure 1 animals-13-00073-f001:**
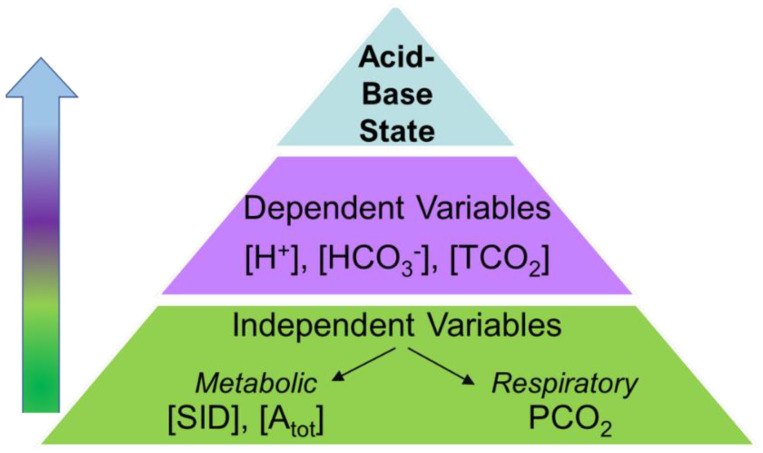
Schematic of the physicochemical approach to acid-base status. The independent variables [SID] (strong ion difference), the PCO_2_ (partial pressure of carbon dioxide) and [Atot] (the total weak acid concentration) determine the concentrations of the dependent acid-base variables ([H^+^], [HCO_3_^−^], and total carbon dioxide concentration ([TCO_2_]). [SID]= [Na^+^] + [K^+^] − [Cl^−^] − [lactate^−^].

**Figure 2 animals-13-00073-f002:**
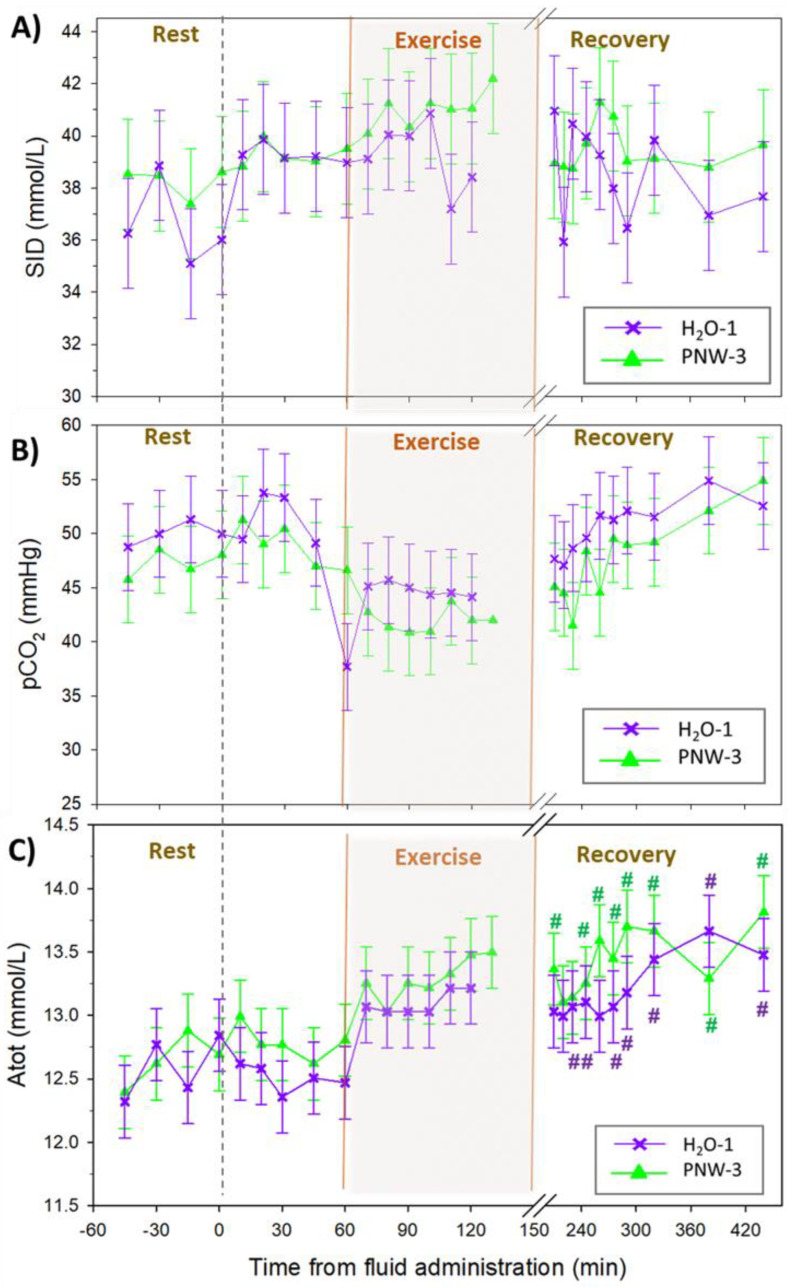
The time courses of plasma independent acid-base variables at rest, during prolonged submaximal exercise, and up to 4 h of recovery. (**A**) Strong Ion Difference [SID], (**B**) partial pressure of carbon dioxide [pCO_2_], and (**C**) total weak acid concentration [Atot] for *n* = 4 horses. In randomized crossover design horses were administered either 3 L of a balanced hypotonic oral electrolyte solution (PNW-3), or 1L of water alone (H_2_O-1), 60 min before they exercised on a treadmill at 30% of peak VO_2_ until voluntary fatigue. Values are mean ± SE. There were no significant differences between groups. # indicates significantly (*p* < 0.05) increased compared to pre-exercise.

**Figure 3 animals-13-00073-f003:**
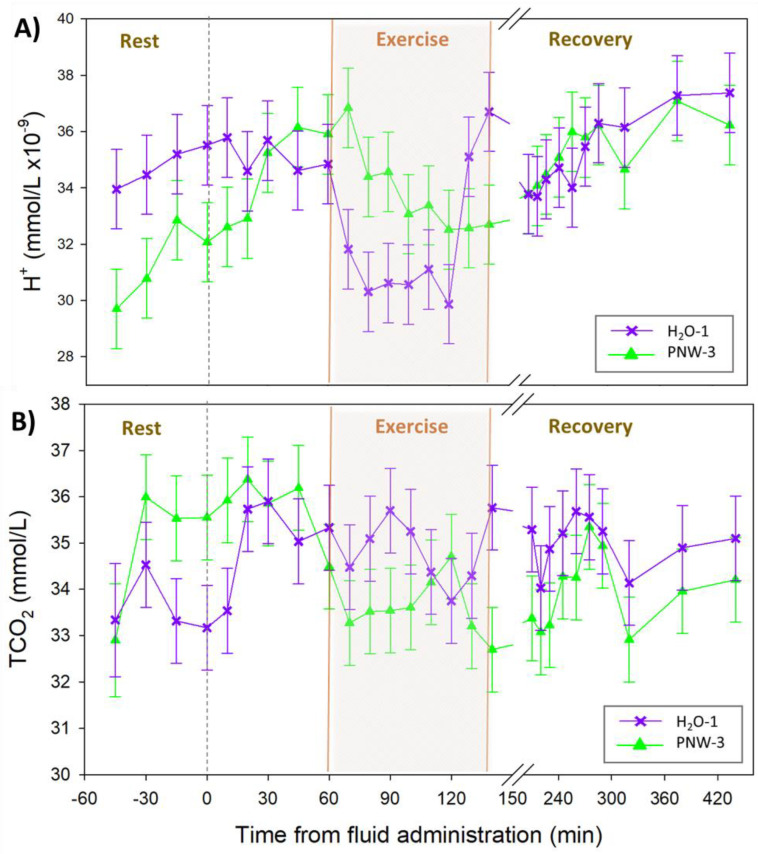
The time courses of plasma dependent acid-base variables at rest, during prolonged submaximal exercise, and up to 4 h of recovery. (**A**) hydrogen ion [H^+^] and (**B**) total carbon dioxide [TCO_2_] for *n* = 4 horses. In randomized crossover design horses were administered either 3 L of a balanced hypotonic oral electrolyte solution (PNW-3), or 1L of water alone (H_2_O-1), 60 min before they exercised on a treadmill at 30% of peak VO_2_ until voluntary fatigue. Values are mean ± SE. There were no significant differences over time or between groups.

**Figure 4 animals-13-00073-f004:**
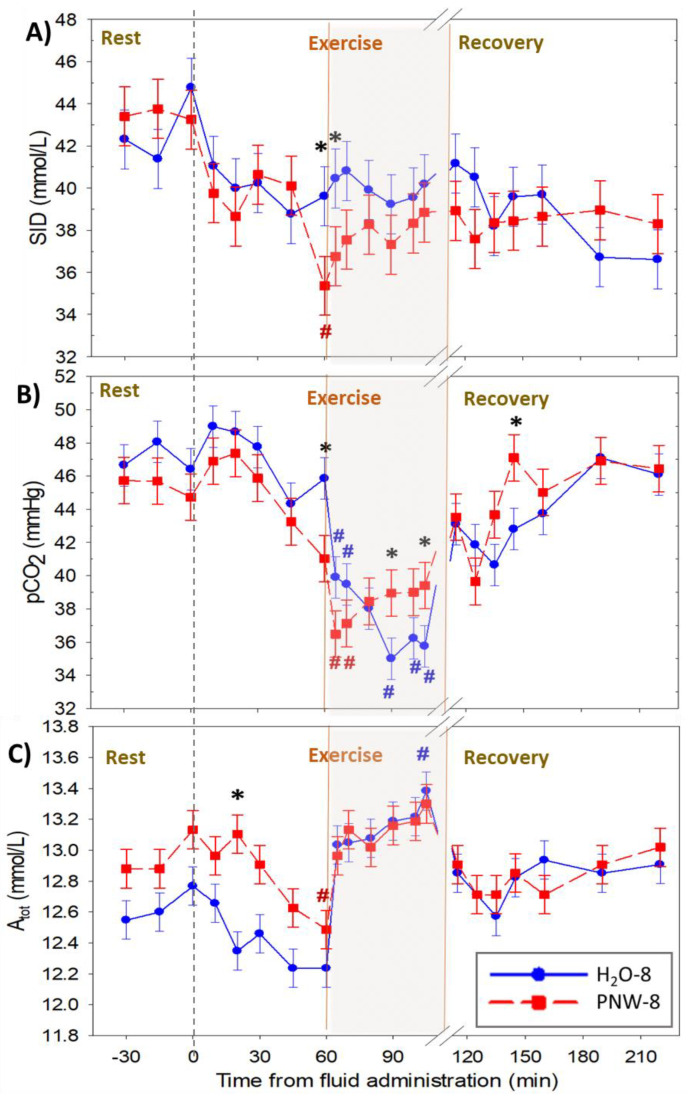
Time course of plasma independent variables at rest during moderate intensity exercise and up to 2 h of recovery. Data show means ± SEM for (**A**) strong ion difference ([SID]), (**B**) partial pressure of carbon dioxide (pCO_2_) and (**C**) total weak acid concentration ([Atot]) for *n* = 4 horses. In a randomized crossover design horses were administered wither 8 L of water (control) or a balanced, hypotonic electrolyte solution (PNW-8) 60 min before they exercised on a treadmill at 50% of peak VO2 for 45 min. # and * denote *p* < 0.05 vs. time 0 and 8 L H_2_O (H_2_O-8), respectively.

**Figure 5 animals-13-00073-f005:**
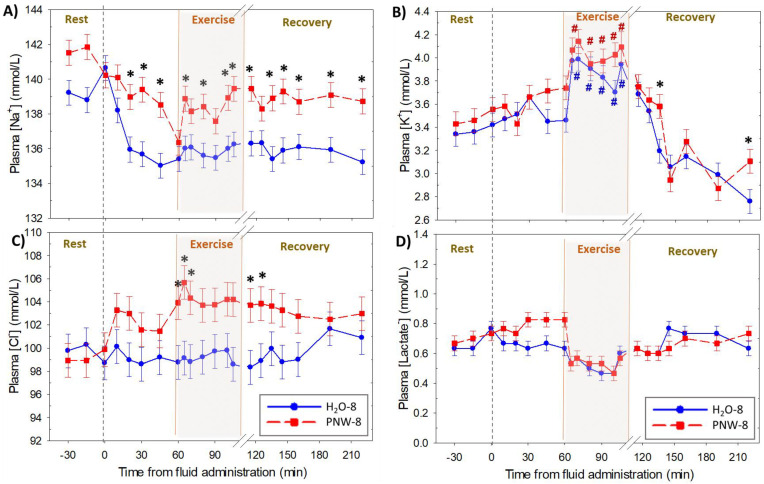
Time course of plasma strong ion concentrations at rest, during moderate intensity exercise and up to 2 h of recovery. Data are mean ± SEM for plasma concentrations of (**A**) sodium (Na^+^), (**B**) potassium (K^+^), (**C**) chloride (Cl^−^) and (**D**) lactate^−^ for *n* = 4 horses. In a randomized crossover design horses were administered wither 8 L of water (control) or a balanced, hypotonic electrolyte solution (PNW-8) 60 min before they exercised on a treadmill at 50% of peak VO_2_ for 45 min. # and * denote *p* < 0.05 vs. time 0 and 8 L H_2_O (H_2_O-8), respectively.

**Figure 6 animals-13-00073-f006:**
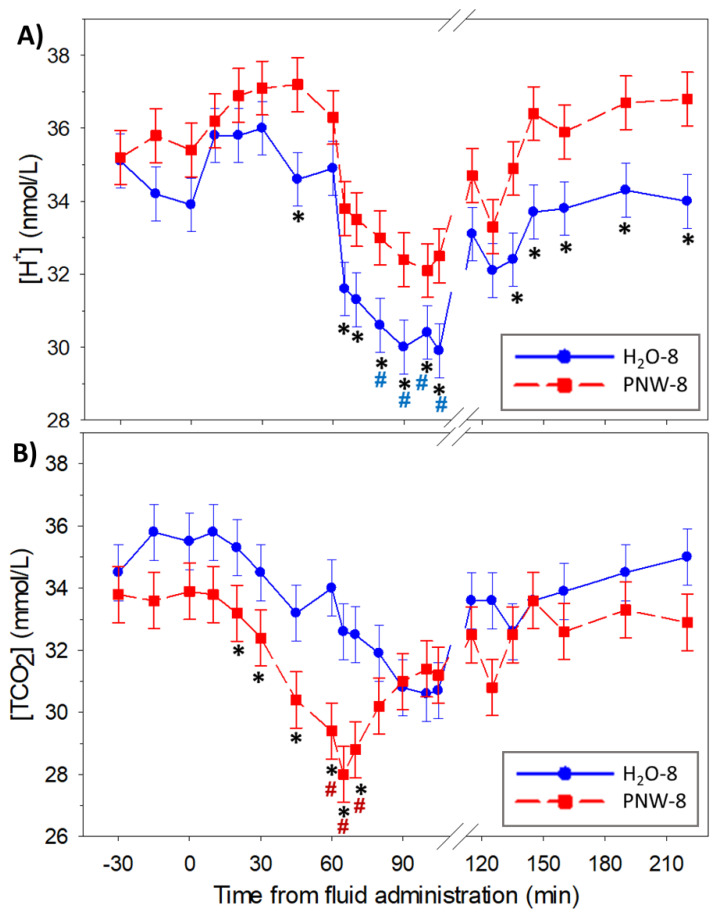
Time course of plasma dependent acid-base variables at rest, during moderate intensity exercise and up to 2 h of recovery. Data are mean ± SEM for plasma concentrations of (**A**) hydrogen ion (H^+^) and (**B**) total carbon dioxide ([TCO_2_]) for *n* = 4 horses. In a randomized crossover design horses were administered wither 8 L of water (control) or a balanced, hypotonic electrolyte solution (PNW-8) 60 min before they exercised on a treadmill at 50% of peak VO_2_ for 45 min. # and * denote *p* < 0.05 vs. time 0 and 8 L H_2_O (H_2_O-8), respectively.

**Figure 7 animals-13-00073-f007:**
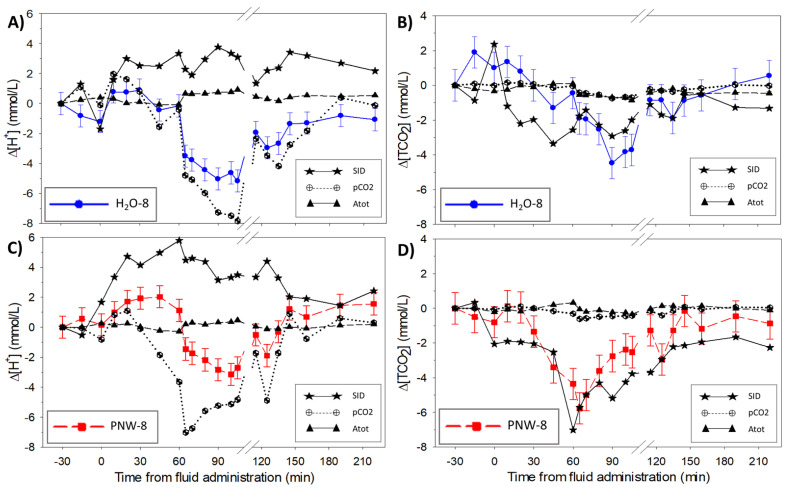
Contribution of the independent variables to the changes in plasma [H^+^] (**A**,**C**) and [TCO_2_] (**B**,**D**). *n* = 4 horses were administered, in a randomized crossover design, 8 L of water as a control (H_2_O-8; A and B) or 8 L of a balanced, hypotonic oral electrolyte solution (PNW-8) 60 min before they exercise on a treadmill at 50% of peak VO_2_ for 45 min. The independent variables contributing to acid-base state are [SID], strong ion difference; pCO_2_, partial pressure of carbon dioxide; and [Atot], total weak acid concentration.

**Figure 8 animals-13-00073-f008:**
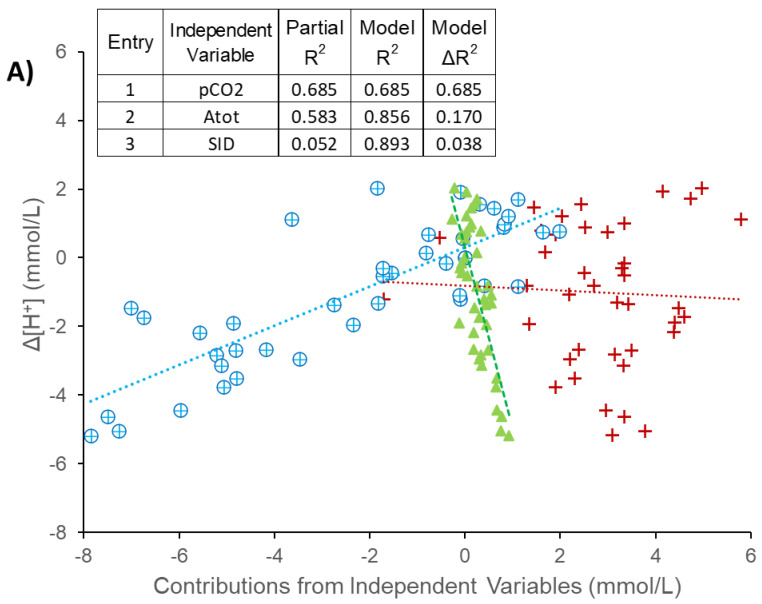
Forward stepwise linear regression analysis confirming the contribution of the independent variables to the changes in plasma dependent variables (**A**) [H^+^] and (**B**) [TCO_2_]. *n* = 4 horses were administered, in a randomized crossover design, 8 L of water as a control (H_2_O-8; A and B) or 8 L of a balanced, hypotonic oral electrolyte solution (PNW-8) 60 min before they exercise on a treadmill at 50% of peak VO_2_ for 45 min. The independent variables contributing to acid-base state are [SID], strong ion difference; pCO_2_, partial pressure of carbon dioxide; and [Atot], total weak acid concentration.

**Table 1 animals-13-00073-t001:** Study design summary.

*Animals*	n = 4 Conditioned Mares	
* **Experimental Series** *	Series 1 Low intensity to voluntary fatigue	Series 2 45′ of moderate intensity
* **Pre-Exercise Supplement** *	1 L H_2_O 3 L PNW	8 L H_2_O 8 L PNW
* **Exercise** * * **Intensity** *	35% peak VO_2_	50% peak VO_2_
* **Exercise Time** *	Voluntary fatigue	45 min

**Table 2 animals-13-00073-t002:** The quantity and measured concentrations of the oral electrolyte supplement (PNW) as measured using a blood gas and ion analyzer (Nova Statprofile 5, Nova Biomedical, Waltham, MA, USA).

	PNW	
	g/L	g/mole
Sodium	1.5	50
Potassium	0.98	25
Chloride	2.7	76.6
calcium	0.27	2.1
Magnesium	0.15	1.2
glucose	5.5	63.3
osmolarity	212 mOsm/kg H_2_O	

**Table 3 animals-13-00073-t003:** Cumulative exercise-induced whole body losses of fluid and ions and the amounts replenished during the Low Intensity Time to Fatigue Trial (top), and Moderate Intensity Standardized 45-min Exercise Trials (bottom). Values are Mean ± SEM for *n* = 4 horses. Cumul = cumulative whole body ion loss over the entire period of sweating (ie; until ~20 min of recovery). Deficit remaining = total body deficit. Ψ, § indicate significantly different from the H_2_O-1 and H_2_O-8 trials, respectively.

Trial	Variables	Fluid (L)	Na^+^	K^+^	Cl^−^
			(mmoles)	(mmoles)	(mmoles)
**H_2_O-1**	Cumul loss	2.7 ± 0.8	280 ± 69	102 ± 30	338 ± 108
	PNW add	1.0	0	0	0
	% replaced	37 ± 11%	----	----	----
	deficit remaining	−0.6%	~1.2%	~0.3%	~2.4%
**PNW-3**	Cumul loss	4.6 ± 1.4 ψ	464 ± 135 ψ	127 ± 39	618 ± 164 ψ
	PNW add	3.0	195.9	81	253.8
	% replaced ψ	65 ± 16%	42 ± 11%	64 ± 18%	41 ± 10%
	deficit remaining	−0.5%	−1.2%	−0.2%	−2.5%
**H_2_O-8**	Cumul loss	10.9 ± 1.8 ψ	863 ± 61 ψ	79 ± 5	1107 ± 77 ψ
	PNW add	8.0	0	0	0
	% replaced	73 ± 12%	----	----	----
	deficit remaining	−1.0%	−4.8% ψ	−1.3% ψ	−9.9% ψ
**PNW-8**	Cumul loss	8.6 ± 3.4 ψ	1213 ± 63 ψ §	196 ± 7 ψ §	1605 ± 133 ψ §
	PNW add	8.0	522	216	677
	% replaced ψ	93 ± 31%	43 ± 2%	110 ± 4%	42 ± 3%
	deficit remaining	−0.2%	−2.2% §	+0.1% §	−4.5% ψ §

## Data Availability

The data that support the findings of this study are available from the corresponding author upon reasonable request.

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
