# Peer review of "Tracing Acid-Base Variables in Exercising Horses: Effects of Pre-Loading Oral Electrolytes"

_animals, 2022, doi:10.3390/ani13010073_

Round 1

Reviewer 1 Report

After reading the article I would like to inform you that I liked the report even though it is simple and very objective, without any big news.

However, I did not understand why the treatments (solutions, volumes, etc) were not challenged in both exercise programs (35% and 50% VOmax). This would be very important for the work as a whole.

All article topics are well-written and clear. I want to suggest more feeding details (concentrate characteristics and type of hay; feeding schedule the day before the exercise challenge), as they themselves point out this can interfere with the final results.

Reviewer 2 Report

Main comments:

The article is very interesting and the results presented are very relevant. Discussion is nice and results have important practical consequences.

The main drawbacks or aspects to be modified are:

- Very limited number of horses, only 4, but this is partly understood by the complication of the studies carried out.

- The material and methods section should be completely rewritten. It is presented in a very confusing way for the reader. Perhaps it would be better to name the series differently, depending on the characteristics of the exercise performed.

- The nomenclature for the different trials changes throughout the text, which makes understanding difficult.

- The authors should consider changing the name of the different experiments, considering that later, due to the lack of significant differences, they jointly process the results derived from two of them.

- In my opinion, only unpublished data should be presented in this manuscript. It is not very clear to me if the authors also have described previously published data (it seems that they did), which does not seem right to me. If so, I would recommend its elimination, making reference to these data when needed to justify the new data provided in this work.

The introduction, results and, above all, the discussion are very well presented, but the material and method section should be rewritten to make it easier to understand the experimental protocol.

Specific comments

Introduction

Line 93. There is an incomplete sentence to remove or to complete

Material and methods:

Table 1. I find confusing to consider series 2 of experiments as a 'standardized test', because it is just a prolonged submaximal (VO2max) constant intensity- exercise. To avoid the confusion of the readers with additional nomenclature, I would recommend using: series 1 and series 2, as used in the text or used a term associated with the type of exercise (indicating percentage of VO2max or exercise to fatigue and 45 min… ).

In addition, it would change the order of the presentation of the table, indicating first the series to which it corresponds, secondly the intensity of the exercise that has been carried out, thirdly the time of exercise and finally what the supplementation consisted of, indicating that they were different trials. Also, I think it would help the reader to indicate in the legend of the table the meaning of the abbreviations. It would be more comfortable, instead of looking for in the text to be able to understand the table.

Table 1. Please indicate somehow that 1l REC is a 1 of electrolytes and not of water.

Line 149. Did the authors present results concerning plasma volume obtained with Evans blue?

Lines 172-174. This information should also be indicated in Table 1 concerning to the characteristics of the experimental protocol.

Sample collection

Lines 177-178. The timing of the sampling is not very clear. They were also taken every 10 min after preloading fluid/electrolytes. It would be easier to understand if a graphical representation of the sample extraction times was made

Sampling times for Evans Blue?

Lines 181 and on. Because the authors have differentiated between 'sample collection' and 'analytical methods and calculations', the information concerning measurement systems (i.e. Nova Statprofile...) should either be introduced in the next section or merge both. The same with respect to the PP measurement.

 RESULTS

I did not find how long was the duration of the exercise on series 1 and if there were significant differences between trials.

Please, try to use the same abbreviations in the text (ex: PNW-1 vs 1L PNW)

I do not understand what H20-1,1 means?, did the authors put together data derived from H2O-1 and REC-1?. In my opinion, the authors should critically review the abbreviations, because it is quite difficult to follow the manuscript because of that.

Lines 216-217. This means that the authors are presenting in Table 3 the results of a previous study?. The authors indicate in the table Cumul. Los, % replaced.. deficit remaining… Nothing in material and methods has been presented on these calculations.

I assume that the authors refer to their previous article (22). They also present data related to sweat (sweat fluid and ion losses), but this has not been described in material and methods. It appears that the authors are presenting data from a previously published article, which does not seem correct to me. On the one hand, all this makes extremely difficult to understand the article. The article should be self-explanatory, without the necessity of having read a previous article. Second, if the data have already been presented previously, they should not be presented again in this manuscript

Lines 222-223. Please redact better this idea. Series 2 is repeated.

Figure 3. In lines 235-236, the authors indicated that there were no significant differences in H+ or TCO2 between both groups of trials (2 trials because the authors mixed data for different trials because significant differences were not previously found). However, when having a look on the figure 3, both in H+ and TCO2 appeared (at least graphically) that some significant differences might happen during exercise. Please, could the authors checked in the figures are well done or try to explain why these differences?

  It appears that in H2O-1,1 trial, during exercise, mean values around 36.5 mmol/ were found (approximate values from the figure) and PNW-1,3, values around 33.5 mmol/l, with a small SD.

DISCUSSION

Nice discussion

Line 306. Here, the authors indicated peak VO2 vs VO2max, before. Because sometimes both terms are no synonymous, the authors should indicate how the VO2max was calculated. Peak VO2 is preferred when VO2max is not reached, but it represents the highest values obtained by the horse (it is probably the value obtained if the horse are not very fit at the moment of the study or other horses different from racehorses). VO2max should be used when a plateau of VO2 is reached. Please, explain that.

Line 351. I am not sure if the authors should use here the term ‘hypoglycemia’ instead of reduction of plasma glucose concentrations. Hypoglycemia should be reserved for these cases when glucose concentrations are lower than the reference values (for the device used if possible). In horses, a hypoglycemia is considered with glucose concentrations are lower than 75 mg/l or 4.2 mmol/l, with small modifications according to the device and horse population used. Please, check if in this case, a reduction of plasma glucose happened or a ‘real’ hypoglycemia.

Lines 352-356. It would be nice if the authors introduce a sentence to specify that the amount of glucose provided in the present manuscript (31 mmol/l) is much lower that this provided in reference 35. 

Lines 400-402. This is a nice appreciation. In fact, many riders/trainers do not use preload of electrolytes under the belief that they will be removed by the horse if an excess exists, but they use with the aim of accelerate thirst, because many horses is competition have lose some thirst. It would be interesting if the authors in the future can scientifically demonstrate how long this ‘storage’ of electrolytes are kept before exercise.

Reviewer 3 Report

General comment: The manuscript “Tracing acid-base variables in exercising horses: effects of pre- loading oral electrolytes “is an interesting paper concerning the evaluation in horses of  the effect of large volume (up to 8 L) pre-exercise electrolyte supplementation on plasma acid-base variables at rest, during moderate intensity exercise and during recovery. The study is a continuation of researches of the same and other authors which are mentioned in the broad and well-structured Introduction. Expanding of known branches of knowledge is always profitable for the science development. The results obtained suggested that pre-loading administration of large volume (8L) electrolyte supplement, compared to water, in order to replace sweat fluid and electrolyte losses, abolished the mild alkalosis that occurred with long-lasting submaximal exercise and resulted in improved acid-base state compared to the alkalosis typical of prolonged exercise. The paper is interesting, because the contents addressed in this study are worthy of further investigations, from both the speculative and the applied points of view. The manuscript is attractive and easy to read and the results obtained are clearly presented and discussed. Hence, I underline that the manuscript is valuable and the results are worth publishing in Animals.

Title: It is suitable and it well describes the experiment presented in the manuscript.

Abstract and Summary: They are suitable. Abstract clearly identifies the interest for this research and its possible relevance. It recaps the information contained in the main text without repetitions.

Introduction: The Introduction provides adequate background, making evidence for the strong interest in acid-base balance within the context of athletic performance. The Introduction also reports the former studies of the same Authors, that confirm that this study is the final part of an extensive series of experiments that traced oral electrolytes in resting and exercising horses, from gastric emptying and intestinal absorption through to their appearance in plasma and ultimately uptake into cells and tissues. This section is clear, well-constructed, although concise; it includes some specific literature references that make evident the interest of the paper in the field of basic, applied and clinical physiology, clearly explaining the aim of the study.

Materials and Methods: This section is adequate, clearly reporting the information about animals used in the study, the study design, the experimental procedures, the samples’ collection, the analytical methods and calculations adopted and the data evaluation. The section makes evident the main concern of the study design, due to the limited number of horses used in the study and to their different stress conditions, as well as to the lack of measurement of blood-gas and acid-base parameters at a standardized temperature. Nevertheless, these failures of the study have been correctly taken into consideration and well discussed later and further on. I suggest to add more information about data concerning the horses involved in the study (e.g., training level, previous experience), and the specific activities usually carried out. Age, in particular, and previous experience of horses could largely influence the results. In the same time, the environmental conditions (presence of co-specific, presence of stranger people) during activity were not considered and further information could be valuable.

Results: The results are clear, well exposed clearly and logically, and the data are clearly presented in the figures and tables.

Discussion: The discussion is well organized and balanced. The comments reported in discussion are pertinent to the results achieved. The authors critically examine their results in the light of the state of science highlighted in the introduction and former practical experience. The discussion of results is extensive and clear.  Discussion follows a logical line, and conclusions are drawn from the study related to the aim of the study and potentially plausible in terms of the results obtained and applied in equine sport performance.  The interpretation of results proposed by the authors in the discussion could be shared. The paper offers the perspective for further study.

References: They are appropriate and mainly up-to-date. I suggest to add the term “in press” to citation No. 4.

Tables and Figures: They are clear and explicative.

Decision: The current manuscript is acceptable for publication after minor revision.

Round 2

Reviewer 2 Report

The authors have addressed all the questions proposed. It has been greatly improved. 

The article is OK for me now